# Assessment of DNA damage of smooth muscle cells in tunica media of human arterial allografts using Comet assay method

Miroslava Jandová[1,2]*, Alexander Pilin[3], Ivan Mazura[3], Radka Lainková[4], Ivan Matia[5], Myroslav Salmay[4], Pavel Měřička[1], Ondřej Pecha[6], Libor Janoušek[7], Tomáš Grus[4], Rudolf Špunda[4], Jaroslav Lindner[4], Dana Čížková[2], Jiří Záhora[8], Miroslav Špaček[4]

1 Tissue Bank, University Hospital Hradec Králové, Hradec Králové, Czech Republic, 2 Department of Histology and Embryology, Faculty of Medicine in Hradec Králové, Charles University, Hradec Králové, Czech Republic, 3 Institute of Forensic Medicine and Toxicology, General University Hospital and First Faculty of Medicine, Charles University, Prague, Czech Republic, 4 2nd Dept. of Surgery-Dept. of Cardiovascular Surgery, First Faculty of Medicine, Charles University and General University Hospital, Prague, Czech Republic, 5 Department of Cardio-Vascular Surgery, WIGEV - Clinic Floridsdorf and Karl Landsteiner Institute for Cardio-Vascular Research, Vienna, Austria, 6 Technology Centre of the Czech Academy of Sciences, Department of Strategic Studies, Prague, Czech Republic, 7 Department of Transplantation Surgery, Institute for Clinical and Experimental Medicine, Prague and First Faculty of Medcine, Charles University, Prague, Czech Republic, 8 Department of Medical Biophysics, Faculty of Medicine in Hradec Králové, Charles University, Hradec Králové, Czech Republic

* miroslava.jandova@fnhk.cz

## Abstract

During the harvest and subsequent processing of blood vessels for allogeneic transplantation, DNA damage can occur. A suitable method for quantifying this damage may be the comet assay, also known as the single cell gel electrophoresis assay, which is used to measure DNA damage at the single-cell level. We evaluated the key indicators of muscle cell DNA damage in the *tunica media* of arteries by using the comet assay with four groups: (1) cryopreserved grafts thawed slowly, (2) cryopreserved grafts thawed rapidly, (3) cold-preserved "fresh" grafts harvested as part of a multiorgan procurement, and (4) cadaveric arterial graft harvested at autopsy. Kruskal-Wallis multiple-comparison testing identified a statistically significant difference between rapidly thawed grafts and multiorgan harvested fresh grafts. Evans' correlation criterion indicated a moderately strong correlation between the percentage of DNA in the comet head and warm ischemia duration in the group of slowly thawed grafts, and between the percentage of DNA in the comet head and cold ischemia duration in the group of cadaveric grafts. No association of come assay parameters with age was demonstrated. Comet assay showed that the cryopreservation and storage processes described in this study did not lower the DNA content in comparison with fresh grafts collected during multiorgan harvests in operating rooms and that DNA content was not influenced by the type of thawing protocol.

**Data availability statement:** The data that support the findings of this study are available as Supporting Information files.

**Funding:** This project was supported by the Ministry of Health of the Czech Republic MH CZ-DRO (VFN, 00064165) and by MH CZ-DRO (UHHK, 00179906). The funders had no role in study design, data collection and analysis, decision to publish, or preparation of the manuscript.

**Competing interests:** The authors have declared that no competing interests exist.

## 1. Introduction

The use of allografts for the treatment of vascular pathologies dates back to the 1950s [1–3]. Their use was terminated in the early 1960s, however, due to frequent graft rejection caused by the unavailability of immunosuppression [4,5]. Both fresh and cryopreserved vascular allografts are now used to replace infected vascular prosthesis and in case of critical limb ischemia. First attempts to cryopreserve blood vessels were not successful, as cases of early rupture and degeneration were described [6]. The phenomenon of early arterial graft rupture was explained by David Pegg, who identified microfractures in cryopreserved rabbit arteries caused by devitrification taking place during rapid thawing [7]. Subsequently, improved methods of cryopreservation and decontamination resulted in better graft survival, namely regarding the viscoelastic and inertial properties of the arterial wall [8,9]. Cryopreserved allografts have been used successfully introduced to treat patients with major peripheral vascular infections, greatly reducing operative mortality as well the need for distal amputation [9]. Kieffer *et al.* found that rare specific complications, including early or late allograft rupture and late aortic dilatation, were significantly reduced by using cryopreserved rather than fresh allografts [10]. Autologous veins, antibiotics, or silver-coated prostheses are offered as alternative treatment methods. However, these techniques are associated with complications such as graft failure, hemorrhage, thrombosis, and infection recurrence [8].

A national vascular allotransplantation program was started in the Czech Republic in 2013. It is based on cooperation among specialized organ or vascular transplantation centers interested in transplantation of vascular tissue and licensed as procurement establishments by the national competent authority (State Institute for Drug Control) with the licensed tissue establishment registered in the Compendium of European Tissue Establishment under code CZ000427. The grafts are collected exclusively from multiorgan donors. The program uses a common immunosuppression protocol described in a previous publication [11]. The aim of this program is to ensure a sufficient supply of cryopreserved allografts. Establishing a sufficient and continuously renewed stock of arterial and venous grafts originating from donors with different blood groups and with long shelf life is a prerequisite for achieving this aim. While our approach is based on tissue procurement after minimal warm ischemia during multiple organ and tissue harvest in operating rooms, several cardiovascular tissue banks [12] also practice vascular tissue harvest at autopsies performed usually within 24 hours of the donor's and followed by hypothermic storage in antibiotic cocktails until use as fresh grafts or until cryopreservation.

For the assessment of damage prior to cryopreservation and after thawing, a combination of morphological and functional methods is recommended. Some of them require previous disintegration of tissue (e.g., if flow cytometry is used), while others can be applied *in situ* [13]. Besides assessment of the cell viability of individual vessel wall layers, preservation of basement membrane structure is of great importance for reendothelization of the internal surface of the graft by the cells of the host and for the prevention of thrombosis after transplantation. The basic methodology for displaying individual layers of the vessel, i.e., *the tunica intima* incl. the *endothelium, the*

tunica media, and the adventitia, is the use of routine histological techniques, such as formalin-fixed paraffin-embedded tissue sections stained with hematoxylin and eosin or some other staining method. Methods detecting morphological signs of cell death can also be used. One of these methods is light microscopy, which reveals features characteristic of different cell death types, primarily apoptosis and necrosis. Cell shrinkage, the formation of membrane vesicles, and the formation of apoptotic bodies are observable in apoptosis. Necrotic cells show disrupted plasmalemma, karyolysis, and translucent (eosinophilic) cytoplasm. More detailed information about the morphological changes during cell death are provided by transmission electron microscopy revealing cellular ultrastructure or scanning electron microscopy that exposes the surface of cells, demonstrating, e.g., the formation of membrane vesicles [14,15] or damage to the endothelial layer and basement membrane structures [16]. By immunohistochemistry, specific markers of a certain type of cell death can be detected, e.g., cleaved caspase 3 or cleaved cytokeratin 18 in apoptosis. Accurate quantification methods are used to evaluate mainly apoptosis. Flow cytometry enables the measurement of properties of cells due to the impact of the laser beam on the cell and its subsequent distraction. Based on the scattering parameters, apoptotic cells can be distinguished from each other and from vital and necrotic cells [15]. DNA fragments in cells undergoing apoptosis can be detected by several gel electrophoresis methods [17]. Terminal deoxynucleotidyl transferase-mediated deoxyuridine triphosphate nick-end labeling assay and in situ end labeling are methods that indicate DNA breaks occurring during apoptosis. These methods allow the evaluation of damaged DNA using light microscopy, fluorescence microscopy, or flow cytometry [18,19]. Another method is to determine the mitochondrial membrane potential, when dysfunctional mitochondria is usually connected with a reduction in membrane potential [20]. Caspases play an important role in apoptosis, and their catalytic activity can be determined, e.g., fluorometrically, colorimetrically, or immunochemically (e.g., by immunoblotting) [17,21].

Another approach is based on DNA degradation assessment and is used routinely in forensic medicine for estimation of the post-mortem interval, such as assessing DNA degradation by DNA amplification analysis, restriction fragment length polymorphism analysis, DNA flow cytometry analysis, and the image analysis technique known as the comet assay or single cell gel electrophoresis.

The aim of our study was to evaluate the degree of DNA damage to cells in the tunica media of blood vessels harvested during multiorgan procurement or during autopsy by using the comet assay method. This method allows one to measure both DNA damage and DNA repair in cells. The suspension of obtained cells is applied to low-solidifying agarose and then lysis is performed to release the nuclei, followed by electrophoresis. The negatively charged DNA migrates to the anode depending on the degree of fragmentation, forming comet-like shapes that can be visualized using fluorescence microscopy [22–24]. The parameters used to assess DNA fragmentation are head and tail area, head and tail DNA content (%), tail moment, and Olive moment [25]. The tail moment describes the product of multiplying the length of the comet tail with the % of DNA in the tail, while the Olive moment is the product of the total % of DNA in the tail and the distance between the centers of the mass of the head and tail regions [22].

## 2. Methods

### 2.1 Design of the experiment

DNA damage of smooth muscle cells in the tunica media of arterial allografts harvested from deceased donors was assessed using the comet assay method. The study was conducted between March 23 and December 1, 2021, with approval from the Ethics Committee of the General University Hospital in Prague (Reference no. 1/20 Grant VFN IGP). The study involved tissue samples obtained post-mortem from cadaveric donors. In accordance with applicable regulations, no informed consent was required from the donors' families. Some authors involved in the study had access to identifiable donor information in their role as treating physicians responsible for organ and tissue procurement. However, all data used for research purposes were anonymized prior to analysis, and the Ethics Committee waived the requirement for informed consent for the use of these samples in this study. The rate of DNA damage was evaluated in four groups of allografts: (1) CAG/S – cryopreserved grafts, slowly thawed, (2) CAG/R – cryopreserved grafts, rapidly

thawed, (3) FAG – cold-stored ("fresh") arterial grafts obtained during multiorgan harvest, and (4) CADAG – cadaveric arterial grafts harvested during autopsy. The term "cryopreserved" grafts means that these tissues were subsequently cryopreserved to liquid nitrogen vapor temperatures after collection. The term "fresh" grafts refers to tissues that were not cryopreserved to liquid nitrogen vapor temperatures after collection, but were processed directly for comet assay analysis. The design of the experiment is presented in Fig 1. The evaluated vessels were harvested as part of a multiorgan procurement (groups 1, 2, and 3) or during autopsy (group 4). The vessels of groups 1 and 2 were harvested within multiorgan procurement, subsequently cryopreserved, and thawed using one of two protocols (slowly or rapidly). Group 3's grafts were analyzed fresh after hypothermic storage in the organ preservation solution Custodiol CE (Dr. Köhler Chemie GmbH, Bensheim, Germany) at $T = 4 - 8$ °C. This group represents the control pre-process group for cryopreserved grafts as well as for grafts harvested at autopsy and analyzed fresh after hypothermic storage under the same conditions.

## 2.2 Cryopreservation and thawing method for grafts from multiorgan harvest

Cryopreservation was performed in a cleanroom at purity class A with a class B background using CE-certified 10% dimethyl sulfoxide (WAK Chemie GmbH, Steinbach, Germany), 5% hydroxyethyl starch (Fresenius Kabi, GmbH, Bad Hamburg, Germany), and human serum albumin (Albuminum Humanum, Grifols, Barcelona, Spain). Grafts were frozen in a programmable freezer (Planer Biomed, Sunburry on Thames, England) with a cooling rate of 1 °C min$^{-1}$ down to −90 °C and then 5 °C min$^{-1}$ to −150 °C, and then stored in a vapor phase of liquid nitrogen in a cryo-container (Kryo CE, Taylor Wharton, Milstedt, Germany). Thawing of the grafts was done either slowly (within 2 hours in a refrigerator, $T = 4 - 8$ °C) or rapidly (approx. 7 minutes).

## 2.3 Preparation of samples for comet analysis

Vessels were first cut and thoroughly rinsed in Hank's balanced solution (HBSS). The *adventitia* was carefully removed, and a few muscle fibers from the middle layer of vessels' *media* were taken for processing. Muscle fibers were incubated for 30 minutes in 3 mL of HBSS with $Ca^{2+}$ and $Mg^{2+}$ with the addition of collagenase I (Sigma-Aldrich, PN SCR103) in a ratio of 1:2 at a temperature of 37 °C. After digestion, visible remnants of the connective tissue were removed, and the cell suspension was centrifuged for 15 minutes at 4000 rpm using a minicentrifuge (MiniFuge GDC006, Auxilab S.L., Spain). The incubation solution was aspirated, and the sedimented cells were rinsed twice with HBSS and centrifuged again, this time for 15 and 5 minutes at 4000 rpm (Minifuge). After the HBSS was thoroughly aspirated after the last centrifugation, 10 μL of phosphate-buffered saline (Roche Ltd., Prague, CZ) and 40 μL of low-melting agarose (LMA, Sigma-Aldrich PN A9414) were added to the cell pellet. The suspension was gently mixed, and 2x 5 μL of this mixture was then dropped

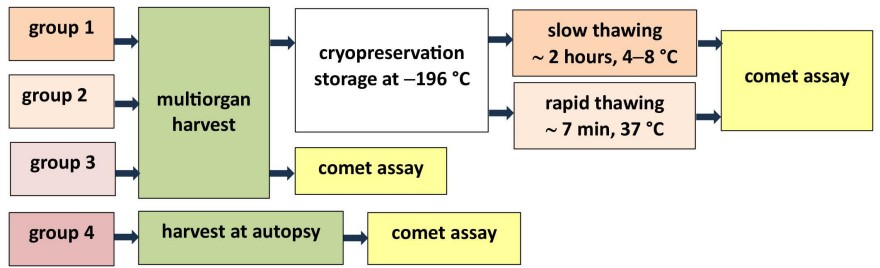

**Fig 1. Design of the experiment.** The figure shows the division of the experiment into 4 groups in terms of the collection environment (multiorgan harvest in the operating room or collection in the autopsy room during the autopsy of a deceased donor) and subsequent processing of the collected arteries.

onto normal melting agarose (NMA)-coated microscope slide prewarmed to 37 °C (Sigma-Aldrich PN A9539). Process flow diagram of the comet assay including preparation of a vascular tissue sample for analysis is presented in Fig 2.

## 2.4 Comet assay

The prepared suspension consisting of myocytes was processed by the alkaline comet assay described Pilin et al., 2024 [26]. Here we present very briefly the procedure: after the cell suspension mixture was dropped in LMA on microscope slides, the suspension was covered with coverslips, the preparation was left on ice, and after the LMA had solidified, the coverslips were torn off, and the preparation was placed in histological cuvettes with lysis solution for 1 hour at a temperature of 4 °C. The end of lysis was followed by the unfolding phase of biological structures in electrophoretic buffer for 20 minutes at laboratory temperature. Electrophoresis was performed in a BIO-RAD Sub-Cell GT, PowerPac, at 25 volts, 300 mA for 20 minutes. Due to the heating of the electrolyte (avoidance of gradual heating of the LMA), electrophoresis was performed in a refrigerator at a temperature of 4 °C. After electrophoresis, the gels were briefly rinsed in neutralization buffer and stained with 1–2 μL of SYBR Gold staining solution (SYBR Gold Nucleic Acid, 10,000 × concentrate in DMSO, Thermo Fisher Scientific PN: S11494). The resulting comets were photographed with a Lucia QI825 camera and evaluated

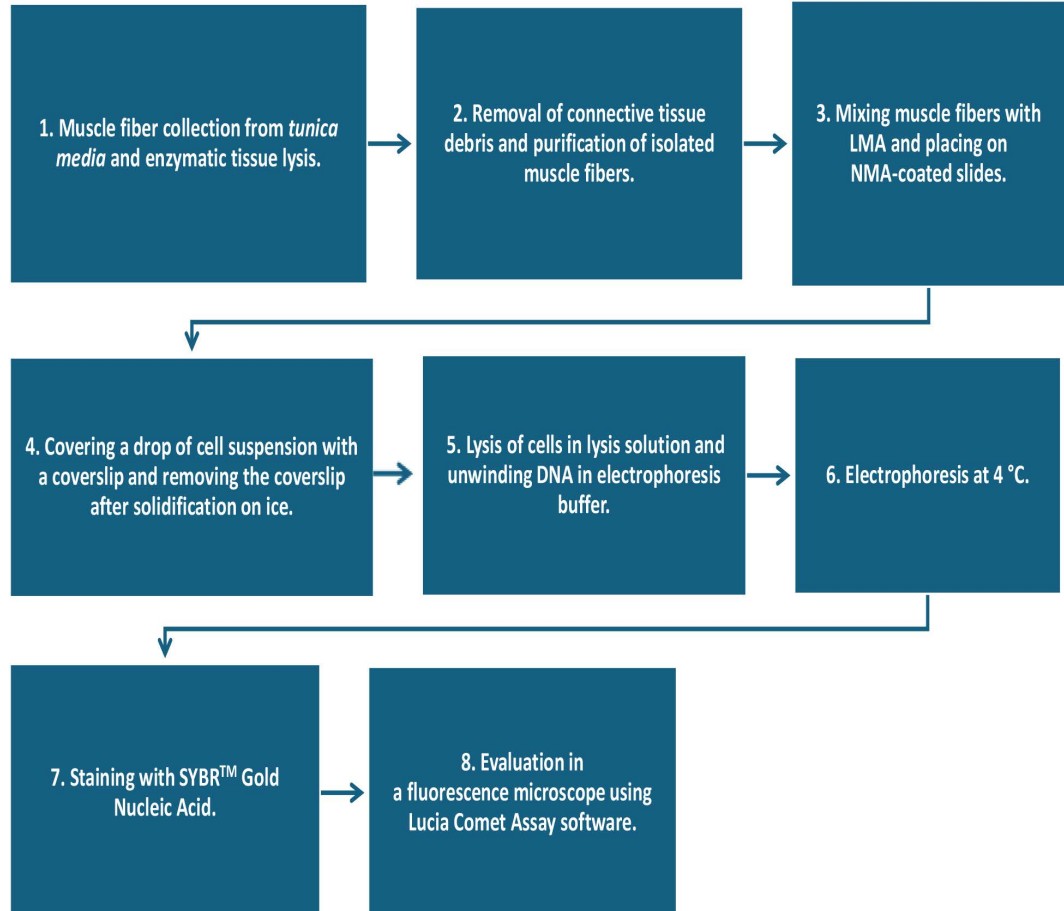

**Fig 2. Process flow diagram of the comet assay including preparation of a vascular tissue sample for analysis.** LMA denotes low-melting agarose; NMA denotes normal melting agarose.

with Lucia Comet Assay Analysis software (Laboratory Imaging s.r.o., Prague) in an Axiomat fluorescence microscope (Carl Zeiss AG, Oberkochen, Germany) with a 450–490/520 nm excitation/emission filter at 200× magnification. We evaluated the results with the following parameters: *head DNA (%)*, *head radius, head area, tail DNA (%), tail length, tail area, sum intensity, tail moment,* and *Olive moment*. A comet consists of two parts: a "head" that contains undamaged or long strands of DNA and a "tail" that contains damaged DNA of different lengths depending on the extent of damage [27]. The *tail moment* describes the product of multiplying the length of the comet tail with the percentage of DNA in the tail, while the *Olive moment* is the product of the total percentage of DNA in the tail and the distance between the centers of the masses of both head and tail regions. The *Olive moment* is particularly useful to describe heterogeneity within a cell population, as it picks up variations in how the DNA is distributed in the tail [22].

## 2.5 Statistical analysis

The data were statistically evaluated using MS Excel 2016 (Microsoft Corp., Redmond, WA, USA), NCSS 10 statistical software, 2015 (NCSS, LLC, Kaysville, UT, USA), available online: ncss.com/software/ncss (accessed April 21, 2023), and TIBCO Statistica 10, 2010 (StatSoft, Inc., Fort Lauderdale, FL, USA), available online: https://www.spotfire.com/contact-us (accessed on March 7, 2025). Since the data did not show a normal distribution, they were described using the median and the first and third quartiles (Q1;Q3). To evaluate the degree of DNA damage within the individual regimes of blood vessel harvest and subsequent processing, we first used Kruskal-Wallis one-way ANOVA at $\alpha = 0.05$ followed by the Kruskal-Wallis multiple-comparison z-value test (Dunn's test) with Bonferroni correction. Outliers were defined as 1.5x the interquartile range (Q3 − Q1), all values, including outliers, were included in the statistical processing.

Correlations between individual comet assay parameters and warm/cold ischemia time, donor´s age and storage time (grafts stored in liquid nitrogen vapor) were determined using Spearman's test at $\alpha = 0.05$. The strength of a correlation was determined based on the value of the correlation coefficient according to Evans' criterion for Pearson´s correlation coefficient values [28]: 0.00 − 0.19, very weak correlation; 0.20 − 0.39, weak correlation; 0.40 − 0.59, moderately strong correlation; 0.60 − 0.79, strong correlation; and 0.80 − 1.00, very strong correlation.

## 3. Results

### 3.1 Number of transplanted vascular allografts in the Czech Republic

The numbers of fresh and cryopreserved allografts implanted annually in the Czech Republic in the last 5 years are shown in Fig 3. Source data on the numbers of fresh and cryopreserved grafts for the period 2020–2024 were provided to the authors by the Transplant Coordination Center in Prague, which is an organizational unit of the state under the direct management of the Ministry of Health of the Czech Republic. This center is responsible for the management and administration of transplant registries.

### 3.2 Characteristics of donors and methods

Tables 1–4 show the characteristics of donors, the type of harvested arteries, and the warm and cold ischemia times. Warm ischemia is defined as time from the cessation of blood circulation till immersion of the collected tissue in pre-cooled preservation solution (Custodiol CE, Dr. Köhler Chemie GmbH, Bensheim, Germany) and cold ischemia as time from immersion of the collected graft into precooled solution till start of processing in the laboratory. In the case of cryopreserved grafts, we refer to cold ischemia I (time from immersion of the collected graft into pre-cooled solution till start of processing in the laboratory) and cold ischemia II (time from start of thawing the grafts and their placement in Custodiol CE solution until their processing in the laboratory). As can be seen in Table 4, cadaveric grafts had a more extended time of warm ischemia than other groups.

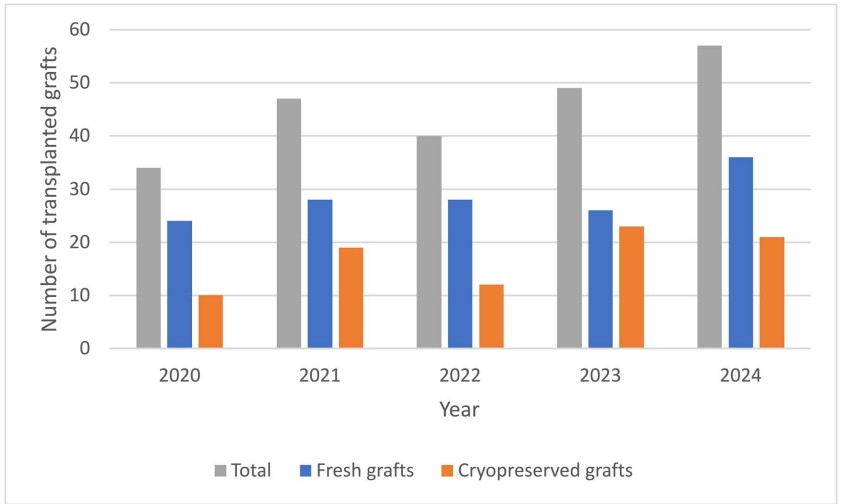

**Fig 3. Numbers of vascular grafts implanted annually in the "fresh" and "cryopreserved" modes in the Czech Republic for the period 2020-2024.** Data were provided by the Transplant Coordination Center in Prague, Czech Republic.

**Table 1. Characteristics of donors, types of arteries harvested during multiorgan procurement followed by cryopreservation, storage in a vapor phase of liquid nitrogen, and slow thawing, and warm and cold ischemia durations.**

| Graft no. | Donor sex | Donor age (years) | Type of harvested graft | Warm ischemia (min) | Cold ischemia I (min) | Cold ischemia II (min) |
|---|---|---|---|---|---|---|
| 4 | male | 49 | *thoracic aorta* | 60 | 1532 | 540 |
| 7 | female | 56 | *thoracic aorta* | 60 | 1589 | 305 |
| 8 | male | 37 | *thoracic aorta* | 95 | 1515 | 315 |
| 11 | male | 27 | *abdominal aorta* | 50 | 710 | 350 |
| 16 | male | 61 | *femoral artery* | 60 | 1342 | 400 |
| Median (Q1;Q3) | | 49 (37;56) | | 60 (60;60) | 1515 (1342;1532) | 350 (315;400) |

Storage period in a vapor phase of liquid nitrogen was median 81.0 (80.5;82.0) months, minimum 80.5 and maximum 82 months, in this group.

**Table 2. Characteristics of donors, types of arteries harvested during multiorgan procurement followed by cryopreservation, storage in a vapor phase of liquid nitrogen and rapid thawing, and warm and cold ischemia durations.**

| Graft no. | Donor sex | Donor age (years) | Type of harvested graft | Warm ischemia (min) | Cold ischemia I (min) | Cold ischemia II (min) |
|---|---|---|---|---|---|---|
| 1 | male | 27 | *thoracic aorta* | 50 | 710 | 410 |
| 2 | male | 58 | *thoracic aorta* | 60 | 1196 | 395 |
| 5 | male | 38 | *thoracic aorta* | 97 | 542 | 194 |
| 6 | male | 33 | *thoracic aorta* | 60 | 938 | 184 |
| 9 | female | 36 | *femoral artery* | 107 | 407 | 228 |
| 13 | male | 40 | *abdominal aorta* | 100 | 930 | 252 |
| 14; | male | 40 | *abdominal aorta* | 100 | 930 | 262 |
| Median (Q1;Q3) | | 38 (34.5;40) | | 97 (60;100) | 820 (626;934) | 252 (211;328.5) |

Storage period in a vapor phase of liquid nitrogen was median 80.0 (76.0;81.0) months, minimum 64 and maximum 87 months, in this group.

**Table 3. Characteristics of donors, warm and cold ischemia durations, and types of arteries harvested during multiorgan procurement followed by comet assay on fresh grafts.**

| Graft no. | Donor sex | Donor age (years) | Type of harvested graft | Warm ischemia (min) | Cold ischemia (min) |
|---|---|---|---|---|---|
| 25 | male | 45 | abdominal aorta | 60 | 738 |
| 26 | male | 57 | abdominal aorta | 60 | 1245 |
| 28 | male | 53 | thoracic aorta | 90 | 135 |
| 29 | male | 53 | iliac artery | 90 | 140 |
| 30 | male | 53 | iliac artery | 90 | 145 |
| 31 | female | 44 | thoracic aorta | 60 | 1263 |
| 32 | female | 44 | femoral artery | 60 | 1268 |
| 34 | male | 76 | thoracic aorta | 50 | 2461 |
| 35 | male | 76 | femoral artery | 50 | 2471 |
| 36 | male | 76 | femoral artery | 50 | 2481 |
| Median (Q1;Q3) | | 53 (47;71.3) | | 60 (52.5;82.5) | 1254 (293.3;2162.8) |

**Table 4. Characteristics of donors, warm and cold ischemia durations, and types of arteries harvested at autopsy followed by comet assay.**

| Graft no. | Donor sex | Donor age (years) | Type of harvested graft | Warm ischemia (min) | Cold ischemia (min) |
|---|---|---|---|---|---|
| 40 | male | 76 | thoracic aorta | 3825 | 1380 |
| 43 | male | 76 | iliac artery | 4625 | 1365 |
| 44 | male | 76 | thoracic aorta | 4635 | 1370 |
| 45 | male | 79 | thoracic aorta | 4530 | 1390 |
| 47 | male | 70 | abdominal aorta | 3465 | 1490 |
| Median (Q1;Q3) | | 76 (76;76) | | 4530 (3825;4625) | 1380 (1370;1390) |

### 3.3 Descriptive statistics

Descriptive statistics showing median (Q1;Q3) values for all variables assessed by comet assay are listed in Table 5. Source data are listed as Supplementary data (S1 − S4 Tables).

### 3.4 DNA damage rate using selected parameters

Data obtained in MS Excel were plotted as median values in all four groups. According to recommendations, we focused on the evaluation of selected parameters [22,25,29]. We present box plots for selected key parameters such as *head DNA (%)* (Fig 4) and *Olive moment* (Fig 5). Fig 4 shows that the largest data variance was seen in the group of grafts cryopreserved with subsequent slow thawing, followed closely cadaveric grafts. The group of rapidly thawed grafts had the most outliers. The statistical significance of the difference ($p = 3.9454$) between the rapidly thawed grafts and the multiorgan harvested fresh grafts was proved using the Kruskal-Wallis multiple-comparison z-value test with Bonferroni correction of α level.

Fig 5 shows the Olive moment box plot. The *Olive moment* is the product of the total percentage of DNA in the tail and the distance between the centers of the masses of both head and tail regions. The graph indicates that in the group of fresh grafts harvested as part of multiorgan donation and in the group of cadaveric grafts, smooth muscle cells are more heterogeneous than cells in other groups. In a Kruskal-Wallis multiple-comparison z-value test with Bonferroni correction of α level, significant differences were observed between the following groups: cryopreserved grafts rapidly thawed and cadaveric grafts ($p = 3.6847$); multiorgan fresh grafts and cryopreserved grafts rapidly thawed ($p = 5.9025$); and multiorgan fresh grafts and cryopreserved grafts slowly thawed ($p = 3.8261$).

**Table 5. Descriptive statistics of the disaggregated data. All variables are represented by values of median (1st quartile;3rd quartile).**

| | All groups | Group 1 | Group 2 | Group 3 | Group 4 |
|---|---|---|---|---|---|
| **Number of grafts** | 27 | 5 | 7 | 10 | 5 |
| **Number of evaluated cells** | 719 | 150 | 148 | 316 | 105 |
| *Head DNA (%)* | 97.8 (52.1;99.4) | 98.8 (32.7;99.4) | 98.9 (96.4;99.5) | 95.7 (53.1;99.3) | 94.1 (37.0;99.6) |
| *Tail DNA (%)* | 2.2 (0.6;47.9) | 1.2 (0.6;67.3) | 1.1 (0.5;3.6) | 4.3 (0.7;46.9) | 5.9 (0.4;63.0) |
| *Sum intensity* | 109.9 (34.5; 419.8) | 30.9 (20.4;40.5) | 41.2 (14.3;75.7) | 383.4 (177.8; 597.0) | 176.7 (101.8;571.2) |
| *Head radius* | 12.0 (8.4;19.7) | 8.1 (7.0;9.6) | 9.2 (6.5;11.1) | 19.4 (14.0;23.2) | 13.8 (11.3;21.0) |
| *Tail length* | 3.2 (0;41.8) | 0.2 (0;37.9) | 0 (0;1.3) | 16.7 (1.5;53.1) | 10.3 (0;78.7) |
| *Tail moment* | 0.1 (0;24.0) | 0.1 (0;27.2) | 0 (0;0.1) | 0.8 (0.1; 27.8) | 1.2 (0;41.1) |
| *Olive moment* | 0.2 (0.1;17.0) | 0.1 (0;18.8) | 0.1 (0;0.2) | 1.2 (0.1; 18.0) | 1.8 (0.1;26.8) |
| *Head area* | 434.6 (225.7;1222.8) | 212.3 (146.7; 271.2) | 272.2 (134.2; 396.5) | 1161.4 (601;1693.9) | 568.4 (379.7;1373.3) |
| *Tail area* | 73.6 (14.3;1039.5) | 18.6 (12.5; 549.2) | 15.5 (10.4;35.1) | 428.4 (36.3;2646.3) | 249.5 (15.7;4488.9) |

Groups: (1) CAG/S – cryopreserved grafts, slowly thawed, (2) CAG/R – cryopreserved grafts, rapidly thawed, (3) FAG – cold-stored ("fresh") arterial grafts obtained during multiorgan harvest, and (4) CADAG – cadaveric arterial grafts harvested during autopsy.

### 3.5 Correlation between individual comet assay parameters and warm and cold ischemia time

Correlations between individual comet assay parameters and ischemia times were determined using Spearman's test. Data of all parameters are listed in Figure 6. Correlation values highlighted in red are significant at $p < 0.05$. According to Evans' criterion [28], the values of the correlation coefficient in Fig 6 show that there was a moderately strong correlation between *head DNA (%)* (and analogously also at *tail DNA percentage*) and warm ischemia time in the group of slowly thawed grafts (Group 1) and between the percentage of DNA in the comet head and cold ischemia time in the group of cadaveric grafts (Group 4). A strong correlation was found between warm ischemia and *tail length*, *tail moment*, and *tail area* in group 1.

### 3.6 Correlation between individual comet assay parameters and donor´s age

A correlation analysis to determine the association of individual comet assay parameters with donor age was performed. In group 1 (cryopreserved grafts, slowly thawed) an association of donor age with the parameters (r – means Pearson´s correlation coefficient): *head DNA (%)* (r=0.69781), *tail DNA (%)* (r= −0.69781), s*um intensity* (r=0.647553), *head radius* (r=0.663311), *Olive moment* (r= −0.677036) and *head area* (r=0.727522) was demonstrated. These correlations were significant at $p < 0.05$. In group 1, the median (Q1;Q3) age was 49 (37;56), which was, for example, a higher value than in group 2 – median (Q1;Q3) was 38 (34.5;40), however, in group 2, the association between comet assay parameters and age was not demonstrated. When performing a correlation analysis on the overall data set, no association of come assay parameters with age was demonstrated. Results of the correlation analysis were added as a supplementary data (S5 − S8 Table).

### 3.7 Correlation between individual comet assay parameters and storage time

A correlation analysis to determine the association of individual comet assay parameters with storage time for cryopreserved grafts stored in liquid nitrogen vapor was performed. In Group 2 (cryopreserved grafts, rapidly thawed) a strong

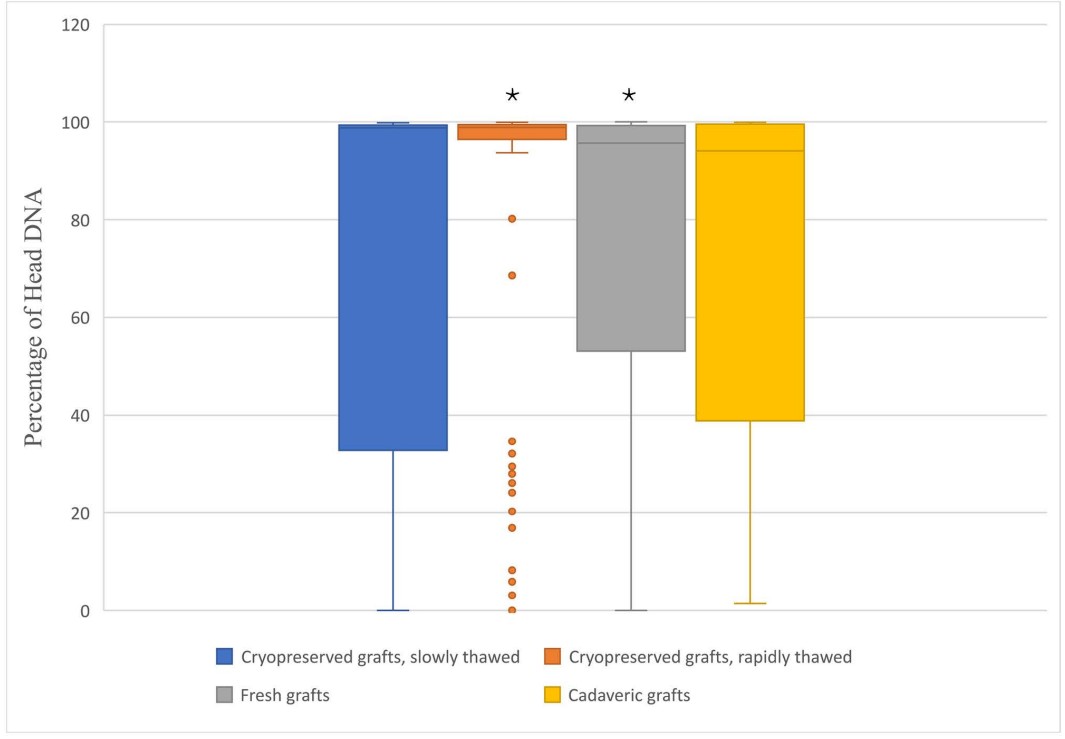

**Fig 4. Box plot shows the percentage of DNA in the head in the comet assay in four groups.** Asterisk indicates groups between which a statistically significant difference (p = 3.9454) was demonstrated using the Kruskal-Wallis multiple-comparison z-value test with Bonferroni correction of α level. The central line within each box represents the median value. The edges of the box indicate the upper and lower quartiles. Whiskers indicate the range of data around the box, without extremes. Outliers were defined as 1.5x the interquartile range (Q3 − Q1). Number of analyzed cells was: 1 – Cryopreserved grafts, slowly thawed: 150; 2 – Cryopreserved grafts, rapidly thawed; 3 – Fresh grafts: 316 and 4 – Cadaveric grafts: 105.

correlation has been shown between storage duration and the following parameters (r – means Pearson´s correlation coefficient): *tail lenght* (r = −0.616574), *tail moment* (r = 0.606510) and *tail area* (r = −0.633326), these correlations were significant at p < 0.05000. Based on these results, it can be assumed that the level of DNA damage will increase with increasing storage time, but in group 1 (cryopreserved grafts, slowly thawed) a strong correlation was not found. Results of the correlation analysis were added as a supplementary data (S5 and S6 Table).

## 4. Discussion

In this article, we present data on fresh and cryopreserved grafts implanted within the last 5 years (Fig 3). Fresh grafts were the most frequently used, as the number of transplanted cryopreserved grafts (about 20 in the last 2 years) indicates, an important role is played by the compatibility between the blood type of the donor and the recipient. In the Czech Republic, there is a possibility to place a patient on waiting list for cryopreserved grafts since 2013. Previously, only fresh vascular grafts were available for transplantation. Before 2013, the average waiting time for obtaining a fresh vascular graft was up to three weeks. During this time, approximately 20% of patients did not receive a graft, with all the consequences that entailed. This was historically one of the main reasons for the creation of the national cryopreserved vascular graft program [30,31]. Since 2013, in most cases, patients placed on the waiting list as urgent recipient, do not have to wait, because there is a supply of grafts in the tissue bank. Therefore, data from the last 12 years are not relevant in this regard. At the same time, it should be noted that centers that also perform organ transplants have better access to fresh grafts due to the multi-organ harvesting they perform, and some surgeons prefer these grafts. Conversely, facilities that

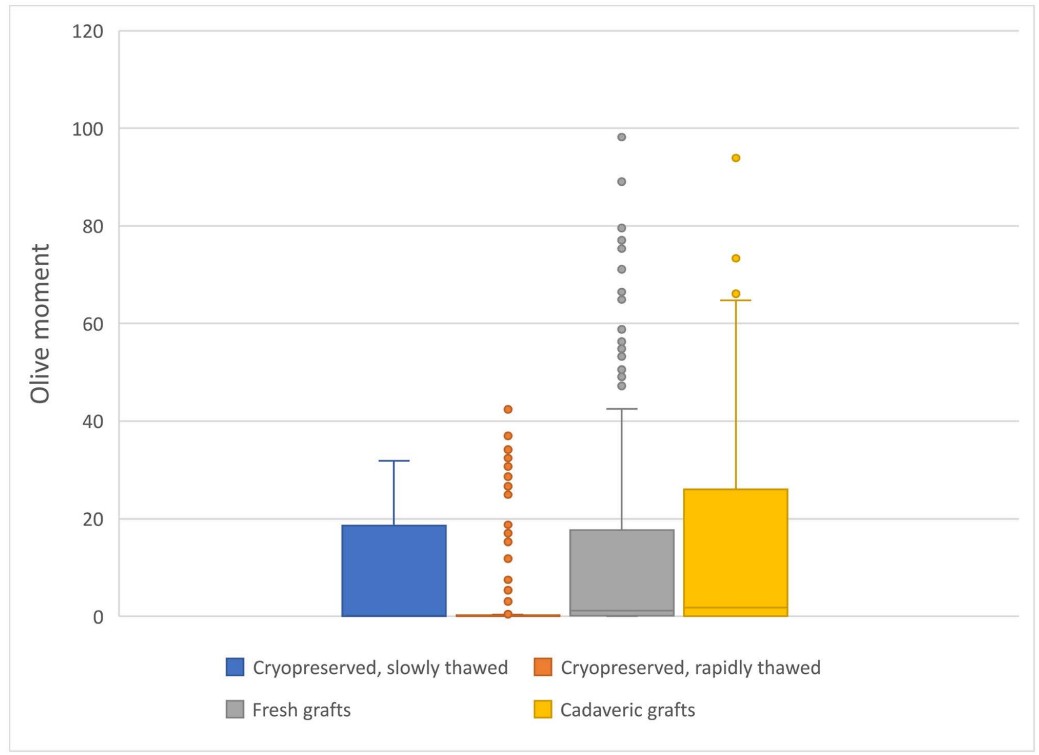

**Fig 5. Box plots showing the *Olive moment* in four groups.** Kruskal-Wallis multiple-comparison z-value test with Bonferroni correction of α level was used. The central line within each box represents the median value. The edges of the box indicate the upper and lower quartiles. Whiskers indicate the range of data around the box, without extremes. Outliers were defined as 1.5x the interquartile range (Q3 − Q1). Number of analyzed cells was: 1 – Cryopreserved grafts, slowly thawed: 150; 2 – Cryopreserved grafts, rapidly thawed; 3 – Fresh grafts: 316 and 4 – Cadaveric grafts: 105.

only transplant vessels more often use cryopreserved grafts. These allow for a semi-elective regimen and are available the day after ordering.

Our cryostability study [32], based on cell viability assessment using fluorescence vital dyes and confocal microscopy, showed immediate post-thaw cell viability of higher than 70% after 5 years of storage. In the present study, we selected the percentage of DNA content in the comet head as the principal parameter for the assessment of cellular damage (Table 5, Fig 4), whereas the sum of the *head DNA (%)* and *tail DNA (%)* is 100%. This parameter can be considered a sufficiently robust marker, together with *tail DNA (%)* [25,29]. The other recommended parameters are *tail length*, *tail DNA (%)*, or *tail moment*; these parameters are recommended for interlaboratory comparison of results [33,34]. The related parameters, such as *sum intensity*, *head radius*, *tail length*, and *head* and *tail area*, characterize the appearance of the comet. The design of the experiment is presented in Fig 1. The original experimental design included 10 samples per group. However, during laboratory processing, some samples were damaged and could no longer be reliably analyzed using the comet assay. These samples were therefore excluded to preserve the quality and validity of the results. As a consequence, the final number of samples differs between groups. Despite this unequal representation, all available samples were processed using the same methodology, and the statistical analysis was performed with respect to the actual number of data points in each group. In total, we analyzed 150, 148, 316, and 105 cells in the individual groups. Although the number of cells varied, the comet assay is highly sensitive to DNA strand breaks, and the number of evaluated cells per group was sufficient to perform reliable statistical comparisons. The unequal numbers resulted from technical limitations (e.g., sample viability after thawing), but all available cells were included to maximize data robustness. Importantly, the observed group sizes still

| Variable | Group 1 | | Group 2 | | Group 3 | | Group 4 | |
|---|---|---|---|---|---|---|---|---|
| | Warm | Cold | Warm | Cold | Warm | Cold | Warm | Cold |
| *Head DNA (%)* | -0,5423 | 0,2943 | -0,2344 | 0,2138 | -0,1029 | 0,1419 | -0,2756 | 0,4324 |
| *Tail DNA (%)* | 0,5423 | -0,2943 | 0,2344 | -0,2138 | 0,1029 | -0,1419 | 0,2756 | -0,4324 |
| *Sum intensity* | 0,0131 | 0,3503 | -0,6329 | 0,3377 | 0,3027 | -0,3062 | 0,0849 | -0,1513 |
| *Head radius* | -0,2454 | 0,2529 | -0,5064 | 0,2253 | 0,2434 | -0,2119 | 0,1163 | -0,1532 |
| *Tail length* | 0,7305 | -0,1742 | -0,2668 | 0,0477 | 0,0899 | -0,1241 | 0,1630 | -0,3107 |
| *Tail moment* | 0,7162 | -0,1822 | -0,2656 | 0,0431 | 0,1057 | -0,1406 | 0,1716 | -0,3204 |
| *Olive moment* | 0,5926 | -0,2931 | 0,1477 | -0,2221 | 0,1136 | -0,1506 | 0,2537 | -0,3960 |
| *Head area* | -0,3353 | 0,2619 | -0,4951 | 0,2451 | 0,2550 | -0,2190 | 0,0941 | -0,1302 |
| *Tail area* | 0,7237 | -0,0375 | -0,4502 | 0,1998 | 0,1301 | -0,1625 | 0,1534 | -0,2900 |

| Pearson's correlation coefficient | Correlation |
|---|---|
| 0.00 - 0.19 | very weak |
| 0.20 - 0.39 | weak |
| 0.40 - 0.59 | moderate |
| 0.60 - 0.79 | strong |
| 0.80 - 1.00 | very strong |

**Fig 6. Heat map showing correlations between comet assay parameters, and warm and cold ischemia times.** Red values are significant at p<0.05. Negative values mean that there is a "the more, the less" relationship between the quantities, i.e., as one quantity increases, the other quantity decreases. The closer the value of the correlation coefficient is to 1 or −1, the stronger the relationship. The strength of a correlation was determined based on the value of the Pearsn´s correlation coefficient according to Evans' criterion. Groups: (1) CAG/S - cryopreserved grafts, slowly thawed, (2) CAG/R - cryopreserved grafts, rapidly thawed, (3) FAG – cold-stored ("fresh") arterial grafts obtained during multiorgan harvest, and (4) CADAG – cadaveric arterial grafts harvested during autopsy.

provided adequate statistical power to detect protocol-related effects on DNA integrity. The power of the test was not calculated because the number of samples was given by realistic possibilities, we presented primary data.

In the Kruskal-Wallis multiple-comparison test, a statistically significant difference was demonstrated only between the groups of grafts obtained during multiorgan harvesting, fresh, and cryopreserved with subsequent rapid thawing (α=0.0125). In all four groups studied, the median DNA content values were above 90%, which implies good DNA preservation. The median content of preserved DNA, i.e., 90% in the head of the comet, is an arbitrarily determined limit. It has been shown that DNA function, i.e., cell vitality and the ability to restart their metabolism after thawing, is closely related to this parameter [32]. The correlation with graft patency and long-term results is probably due more to the extent of preservation of the endothelial layer of the vascular graft after transplantation to the recipient, as well as the preservation of the basal membrane. In such a situation, the layers of the vascular graft media that elicit a strong immune response and contribute to complications such as graft degeneration, dilation, aneurysms, and graft thrombosis in the terms of mid-term patency rates are not exposed to the recipient's immune system and are therefore directly related to the duration of its patency. Resistance to infection is determined more by the biological nature of the graft than by the preservation of its DNA and, in these terms, is not related to the high level of vital DNA preservation. The results of *head DNA (%)* differ in the distribution of values. In case of fresh grafts (Group 3), the median value was 95.7% and the first quartile value 53.1%, while in the cadaveric group (Group 4), the median value was similar at 94.1% but the first quartile value was only 37.0% (Table 5). The difference between these two groups was statistically insignificant, despite the 72-hour interval between the death of the donor and the autopsy performed at the Forensic Medicine Department. However, such a long interval cannot be accepted in practice because of the high risk of unavailability of the hemolysis-free blood sample necessary for obligatory serological testing of donors [35,36]. On the other hand, the cold ischemia time of 24 hours in this study is not sufficient to obtain the results for microbiological swabs taken during the harvest, which are necessary for the release of harvested grafts for clinical application. More data obtained after a longer cold storage period are necessary for more precise comparison of these types of preservation. By contrast, the 24-hour cold ischemia in multiorgan harvest of fresh grafts is fully sufficient if the grafts are used for direct transplantation, preferably within

24 hours. Autopsy-derived tissues often lack standardized procurement protocols, leading to heterogeneity in quality. These grafts are subject to variable ischemic times, temperature fluctuations, and delayed processing, all of which can compromise cell viability and tissue architecture. These factors disproportionately affect grafts with borderline quality, which tend to cluster in the first quartile of outcome distributions. In the Czech Republic only this type of fresh graft transplantation is performed at present. In addition, compared with harvest in the operating room, the environment of the dissection room has a greater risk of microbiological contamination, which may result in the harvested graft being discarded. The discard rate due to contamination [37] is in our practice much lower than described by Jashari *et al*. [38], who use both multiorgan and autopsy harvesting.

Using Evans' criterion for Pearson correlation coefficients, we identified a moderately strong correlation between the percentage of DNA in the comet head and warm ischemia duration in the group of slowly thawed grafts (Group 1), and the percentage of DNA in the comet head and cold ischemia duration in the group of cadaveric grafts (Group 4) (Fig 6). This result was expected.

The finding of a high level of DNA content after cryopreservation and subsequent storage for more than 6 years (median 98.9% after rapid thawing, 98.8% after slow thawing) corresponds with a previously described high cell viability after 5-year storage and represents a complementary cryostability study required by EU legislation [36]. In the case of cryopreserved grafts, the rapidly thawed group (Group 2) showed a very narrow range of values (Q1 = 96.4%) while in the slowly thawed group (Group 1) Q1 value was as low as 32.7%, which was comparable to that of the cadaveric grafts group. This result was again not statistically significant, and it corresponded with results for a comparison of cell viability assessed immediately after thawing in a previous study by the authors [32]. In this study, the statistically significant difference in favor of rapid thawing occurred after 24-hour culture of thawed vessels in the tissue culture medium at 37°C. Such a result was not surprising, as the superiority of rapid thawing is well known [39]. In the cryopreservation of vessels, especially arteries, evidence for the use of slow thawing exists in order to lower the risk of the formation of microfractures during thawing and to prevent early ruptures followed by fatal bleeding as described by Felmer *et al.* [40].

Group 3 also provided a pre-freeze value for comparison with post-thaw values after both protocols. There was again a very minute difference in median DNA content between pre-freeze (95.7%) and post-thaw (98.9% after rapid and 98.8% after slow thawing). The difference was not statistically significant for the slow thawing protocol. For the rapid thawing protocol, there were many outliers in the data set, but overall the data showed the least variance in distribution (Fig 4). When Kruskal-Wallis multiple-comparison test was used, a statistically significant difference was found in favor of post-thaw values ($\alpha = 0.0125$). This improbable result should be verified by other tests, or correction must be made by eliminating pre-freeze values from samples stored for a period above the upper limit of cold ischemia duration in the rapid thawing group (in the rapid thawing group, the Q3 value was 15 hours, while in the multiorgan harvest group it was 36 hours). The best option would be to perform a separate study with known pre-freeze and post-thaw parameters of the same graft. In our previous study [32], we noted that the results of viability assessment in vessel cross-sections were more reliable than the results for assessment of the endothelial layer. The reason is that in the latter case, the most damaged cells may detach from the basement membrane, which may lead to false positive results. This may explain the discrepancy between the results of this study and those of Pilin *et al.* [26], who reported the lowest degree of endothelial damage after cryopreservation followed by slow thawing.

Interestingly, in our study some cryopreserved groups showed higher DNA integrity compared to fresh grafts. Although counterintuitive, similar findings have occasionally been reported in the context of tissue preservation. In the study by Hrubý et al., 2020, the immunogenicity of cryopreserved aortic allografts was significantly lower compared to aortic allografts stored in cold storage [41]. Several factors may account for this observation. First, variability in ischemia duration and post-mortem intervals before sample collection could have affected the extent of DNA degradation in the fresh material. Second, minor differences in handling and processing of individual grafts might have influenced the degree of DNA strand breakage detected by the Comet assay. Finally, it is conceivable that the cryopreservation procedure itself, by rapidly stabilizing the samples at low temperatures, temporarily reduced further DNA damage compared to fresh tissues maintained under non-frozen conditions. These considerations highlight the importance of carefully controlling pre-analytical variables and suggest that the apparent benefit observed in cryopreserved samples should be interpreted with caution. In general,

the results regarding DNA content in the comet head indicated that our cryopreservation, storage, and thawing are well set. Neither this study nor that of Pilin *et al.* included the post-thaw culture period, which may show if the cell and/or DNA damage is reparable or if unrepairable damage results in delayed cell death. The low DNA damage after hypothermic storage and cryopreservation and long-term storage of arterial grafts in liquid nitrogen vapor is evidence of the correct functioning of the standard operating procedures used for tissue collection and cryopreservation and may increase clinician confidence and lead to more frequent application of cryopreserved grafts. In the author's opinion, the results of the comet assay for DNA damage assessment must be considered in the context of the results of other tests based on cell membrane integrity or metabolic activity of cells, which appear to be more sensitive indicators of cell viability.

## 5. Conclusion

The DNA damage assessment using the comet assay showed that the cryopreservation and storage processes described in this study do not lower DNA content in comparison with fresh grafts collected during multiorgan harvests in operating rooms. DNA content was not influenced by the type of thawing protocol; nevertheless, slow thawing is generally preferred for safety reasons and for the prevention of microfractures and early graft ruptures. More data obtained after longer storage are necessary for exact comparison of the quality of grafts collected at autopsy with those gathered in multiorgan harvests and used fresh or cryopreserved. Our long-term experience suggests, however, a lower discard rate due to bacterial contamination in grafts collected during multiorgan harvests than has been reported by centers using autopsy harvest.

## Supporting information

**S1 Table. Comet assay results in group 1 – CAG/S – cryopreserved grafts, slowly thawed.**
(XLSX)

**S2 Table. Comet assay results in group 2 – CAG/R – cryopreserved grafts, rapidly thawed.**
(XLSX)

**S3 Table. Comet assay results in group 3 – FAG – cold-stored (fresh) arterial grafts obtained during multiorgan harvest.**
(XLSX)

**S4 Table. Comet assay results in group 4 – CADAG – cadaveric arterial grafts harvested during autopsy.**
(XLSX)

**S5 Table. Correlations between individual comet assay parameters and donor´s age and storage time in Group 1 – cryopreserved grafts, slowly thawed.**
(XLSX)

**S6 Table. Correlations between individual comet assay parameters and donor´s age and storage time in Group 2 – cryopreserved grafts, rapidly thawed.**
(XLSX)

**S7 Table. Correlations between individual comet assay parameters and donor´s age in Group 3 – fresh grafts obtained during multiorgan harvest.**
(XLSX)

**S8 Table. Correlations between individual comet assay parameters and donor´s age in Group 4 – cadaveric graft harvested during autopsy.**
(XLSX)

## Acknowledgments

The authors would like to thank Scribendi Inc., Canada, for academic proofreading.

## Author contributions

**Conceptualization:** Radka Lainková, Libor Janoušek, Tomáš Grus, Rudolf Špunda, Jaroslav Lindner.

**Data curation:** Radka Lainková.

**Formal analysis:** Alexander Pilin, Ivan Mazura, Ondřej Pecha, Jiří Záhora.

**Funding acquisition:** Miroslav Špaček.

**Investigation:** Miroslav Špaček.

**Methodology:** Myroslav Salmay, Pavel Měřička.

**Supervision:** Ivan Matia, Miroslav Špaček.

**Writing – original draft:** Miroslava Jandová.

**Writing – review & editing:** Dana Čížková.

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
