## [Decision Letter · Decision Letter 0]

8 Aug 2025

Dear Dr. Jandová,

Thank you for submitting your manuscript to PLOS ONE. After careful consideration, we feel that it has merit but does not fully meet PLOS ONE’s publication criteria as it currently stands. Therefore, we invite you to submit a revised version of the manuscript that addresses the points raised during the review process.

We look forward to receiving your revised manuscript.

Kind regards,

Geetika Verma

Academic Editor

PLOS ONE

Journal Requirements: 

 [This project was supported by the Ministry of Health of the Czech Republic MH CZ-DRO (VFN, 00064165) and by MH CZ-DRO (UHHK, 00179906).]. 

4. In the online submission form, you indicated that [The data that support the findings of this study are available from the corresponding author upon the request.].

Reviewers' comments:

Reviewer's Responses to Questions

**Comments to the Author**

1. Is the manuscript technically sound, and do the data support the conclusions?

Reviewer #1: Yes

Reviewer #2: Partly

2. Has the statistical analysis been performed appropriately and rigorously?

Reviewer #1: Yes

Reviewer #2: Yes

3. Have the authors made all data underlying the findings in their manuscript fully available?

Reviewer #1: No

Reviewer #2: Yes

4. Is the manuscript presented in an intelligible fashion and written in standard English?

Reviewer #1: Yes

Reviewer #2: Yes

Reviewer #1: This manuscript presents an assessment of DNA damage in smooth muscle cells from arterial allografts using the comet assay, comparing various graft preservation and thawing strategies. The study is timely and clinically relevant, especially for vascular tissue banking and transplantation, and is based on sound methodology. Overall, the manuscript is of good quality, required minor revision to enhance clarity, interpretability of the study.

Comments

1. How do the study’s findings on DNA damage under various preservation conditions inform future clinical practices or policy guidelines in tissue banks? Could the authors expand the discussion to address the potential impact on graft viability and long-term outcomes?

2. To improve methodological clarity and help readers follow the workflow more easily, a graphical summary—such as a flowchart or schematic—of the sample preparation and comet assay procedure should be included.

3. The authors should clearly state the number of graft samples and cells analyzed per group in the Results section to enhance transparency and support interpretation of the statistical analyses.

4. Given the unexpected observation of higher DNA integrity in some cryopreserved groups compared to fresh grafts, the discussion should address potential factors such as ischemia duration, post-mortem intervals, or variability in sample processing that might explain these results.

5. Figure clarity can be improved by enhancing labelling of box plots (including median lines and whisker descriptions), mentioning sample sizes in figure legends, and adding a visual summary (such as a heatmap or correlation matrix) for correlation results.

6. Authors are encouraged to deposit all the dataset instead of available upon request.

Reviewer #2: The authors leverage a mature network of licensed procurement and tissue establishments, with coordinated multiorgan harvests and rapid cold chain transfer, fully aligned with EU Directives and Czech regulations. These systems enable consistently short cold ischemia times and high-quality cryopreserved vascular grafts. The implementation of slow thawing protocols and comprehensive quality control—including over 70% cell viability post thaw even after extensive storage—demonstrates technical rigor and adherence to best practices in cryostability studies. By comparing fresh vs cryopreserved grafts, including rapid and slow thawing approaches, with consistent metrics such as head DNA% and olive moment, the study provides detailed comparative data across clinically relevant groups

The manuscript presents a valuable overview of Czech Republic's vascular graft transplantation program, combining procedural excellence with cryopreservation validation. However, to strengthen scientific credibility and clinical relevance, the authors may address few concerns. The authors must consider these points critically and revise accordingly.

Major comments

• The manuscript reports or references implantation numbers up to 2016 only. Yet Figures claim to show data for 2020–2024. Please provide the actual source or raw data for allograft implantation numbers during 2020–2024, including methodology for data collection and definitions for “fresh” versus “cryopreserved” categories.

• The sample sizes for some groups (e.g., Group 1 vs Group 2 vs Group 4 in comet assay) appear unequal and small (e.g. 5 vs 7 vs 10 vs 5). Was a priori power calculation performed to justify sample sizes? Please clarify the experimental design and sensitivity to detect effects on DNA integrity across thawing protocols.

• Box plots show substantial variance in slowly-thawed grafts and cadaveric fresh grafts, yet only a specific pairwise comparison achieves statistical significance. How were outliers defined and handled? Was multiple testing correction (e.g. Bonferroni) applied systematically across all pairwise tests? Please provide p values for all comparisons and justification for focus on head DNA%.

• Table 6 indicates moderate to strong correlations between ischemia duration and DNA damage—but only in specific groups (Group 1 and Group 4). Were these associations adjusted for donor age, graft type, cold ischemia I vs II, or storage duration? Please clarify statistical models used, and whether any multivariate analyses were conducted.

• While median head DNA content > 90% suggests good DNA preservation, what thresholds are considered clinically acceptable? Is there known correlation between these comet assay metrics and graft patency, infection resistance, or long term outcomes? Please contextualize DNA damage findings relative to patient outcomes or previous benchmarks.

• The discussion notes there are two waiting lists in the Czech Republic (fresh vs cryopreserved), with surgeon preference guiding assignment. Can the authors provide data on wait times, rejection/discard rates, or post implant outcomes (e.g., graft-related complications), stratified by graft type and thaw protocol?

Minor comments

• Define acronyms clearly at first use (e.g. TE, PE, SOP, Q1/Q3).

• Given the long-term storage (median ≈6–7 years), what is the range of storage durations and did tail moments correlate with time in storage? Could delayed DNA repair post-thaw be assessed?

• Discussion should explicitly address limitations of cryopreserved autopsy-derived grafts despite statistically non-significant median differences—with particular attention to first quartile variance.

• Few typos

**Do you want your identity to be public for this peer review?** For information about this choice, including consent withdrawal, please see our Privacy Policy

Reviewer #1: **Yes: ** Prashant Singh

Reviewer #2: No

---

## [Author Response · Author response to Decision Letter 1]

15 Sep 2025

9th September, 2025

Dear Reviewers,

thank You very much for Your valuable comments that can help me to improve the manuscript.

Please, see my corrections below:

Reply to Reviewer 1:

1. How do the study’s findings on DNA damage under various preservation conditions inform future clinical practices or policy guidelines in tissue banks? Could the authors expand the discussion to address the potential impact on graft viability and long-term outcomes?

Reply: The low DNA damage after hypothermic storage and cryopreservation and long-term storage of arterial grafts in liquid nitrogen vapor is evidence of the correct functioning of the standard operating procedures used for tissue collection and cryopreservation and may increase clinician confidence and lead to more frequent application of cryopreserved grafts. In the author's opinion, the results of the comet assay for DNA damage assessment must be considered in the context of the results of other tests based on cell membrane integrity or metabolic activity of cells, which appear to be more sensitive indicators of cell viability.

(The above text has been added to the Discussion).

2. To improve methodological clarity and help readers follow the workflow more easily, a graphical summary such as a flowchart or schematic of the sample preparation and comet assay procedure should be included.

Reply: The design of the entire experiment is presented in Figure 1. Figure 2 presenting sample preparation and processing for the comet assay has also been added. This figure complements the text part.

3. The authors should clearly state the number of graft samples and cells analyzed per group in the Results section to enhance transparency and support interpretation of the statistical analyses.

Reply: The number of graft samples including the number of evaluated cells is listed in Table 5. The numbers of analyzed cells were also added to the figure legend (Fig. 4 and 5).

4. Given the unexpected observation of higher DNA integrity in some cryopreserved groups compared to fresh grafts, the discussion should address potential factors such as ischemia duration, post-mortem intervals, or variability in sample processing that might explain these results.

Reply: We thank the reviewer for this valuable comment. We agree that the observation of higher DNA integrity in certain cryopreserved groups compared to fresh grafts is unexpected and warrants further discussion. As suggested, several factors may have contributed, including differences in ischemia duration, post-mortem intervals, or subtle variability during sample processing. We have revised the Discussion to address these potential explanations and to emphasize that, while cryopreservation can in some cases stabilize cellular structures, these findings should be interpreted with caution. Additional studies with more controlled conditions will be required to disentangle the influence of these variables. The Discussion part was extended:

,,Interestingly, in our study some cryopreserved groups showed higher DNA integrity compared to fresh grafts. Although counterintuitive, similar findings have occasionally been reported in the context of tissue preservation. In the study by Hrubý et al., 2020, the immunogenicity of cryopreserved aortic allografts was significantly lower compared to aortic allografts stored in cold storage. Several factors may account for this observation. First, variability in ischemia duration and post-mortem intervals before sample collection could have affected the extent of DNA degradation in the fresh material. Second, minor differences in handling and processing of individual grafts might have influenced the degree of DNA strand breakage detected by the Comet assay. Finally, it is conceivable that the cryopreservation procedure itself, by rapidly stabilizing the samples at low temperatures, temporarily reduced further DNA damage compared to fresh tissues maintained under non-frozen conditions. These considerations highlight the importance of carefully controlling pre-analytical variables and suggest that the apparent benefit observed in cryopreserved samples should be interpreted with caution.“

(The above text has been added to the Discussion).

5. Figure clarity can be improved by enhancing labelling of box plots (including median

lines and whisker descriptions), mentioning sample sizes in figure legends, and adding a visual summary (such as a heatmap or correlation matrix) for correlation results.

Reply: The legend of figures 4 and 5 has been modified: ,,Box plot shows the percentage of DNA in the head in the Comet assay. The central line within each box represents the median value. The edges of the box indicate the upper and lower quartiles. Whiskers indicate the range of data around the box, without extremes. Outliers were defined as 1.5x the interquartile range (Q3-Q1). Number of analyzed cells was: 1 - Cryopreserved grafts, slowly thawed: 150; 2 - Cryopreserved grafts, rapidly thawed; 3 – Fresh grafts: 316 and 4 – Cadaveric grafts: 105.“

Table No. 6 has been replaced by Fig. 6 - Heatmap.

6. Authors are encouraged to deposit all the dataset instead of available upon request.

Reply: The raw data and correlation analysis results will be made available as Supplementary data (S1-S8 Tables) of the manuscript - the data file has been uploaded to the Submission system.

Reply to Reviewer 2:

Major comments:

1. The manuscript reports or references implantation numbers up to 2016 only. Yet Figures claim to show data for 2020–2024. Please provide the actual source or raw data for allograft implantation numbers during 2020–2024, including methodology for data collection and definitions for “fresh” versus “cryopreserved” categories.

Reply: Source data on the numbers of fresh and cryopreserved grafts for the period 2020–2024 were provided to the authors by the Transplant Coordination Center in Prague, which is an organizational unit of the state under the direct management of the Ministry of Health of the Czech Republic. This center is responsible for the management and administration of transplant registries. This information has been added to point 3.1 in the Results section.

The term "cryopreserved" grafts means that these tissues were subsequently cryopreserved to liquid nitrogen vapor temperatures after collection. The term "fresh" grafts refers to tissues that were not cryopreserved to liquid nitrogen vapor temperatures after collection, but were processed directly for Comet assay analysis.

The above definitions were added to point 2.1 in the Methods section.

2. The sample sizes for some groups (e.g., Group 1 vs Group 2 vs Group 4 in comet assay) appear unequal and small (e.g. 5 vs 7 vs 10 vs 5). Was a priori power calculation performed to justify sample sizes? Please clarify the experimental design and sensitivity to detect effects on DNA integrity across thawing protocols.

Reply: Thank you for your comment regarding the unequal number of samples across groups and the sensitivity to detect differences in DNA integrity across thawing protocols. The original experimental design included 10 samples per group. However, during laboratory processing, some samples were damaged and could no longer be reliably analyzed using the comet assay. These samples were therefore excluded to preserve the quality and validity of the results. As a consequence, the final number of samples differs between groups. Despite this unequal representation, all available samples were processed using the same methodology, and the statistical analysis was performed with respect to the actual number of data points in each group. In total, we analyzed 150, 148, 316, and 105 cells in the individual groups. Although the number of cells varied, the comet assay is highly sensitive to DNA strand breaks, and the number of evaluated cells per group was sufficient to perform reliable statistical comparisons. The unequal numbers resulted from technical limitations (e.g., sample viability after thawing), but all available cells were included to maximize data robustness. Importantly, the observed group sizes still provided adequate statistical power to detect protocol-related effects on DNA integrity. The power of the test was not calculated because the number of samples was given by realistic possibilities, we present primary data.

(This part was added to Discussion).

3. Box plots show substantial variance in slowly-thawed grafts and cadaveric fresh grafts, yet only a specific pairwise comparison achieves statistical significance. How were outliers defined and handled? Was multiple testing correction (e.g. Bonferroni) applied systematically across all pairwise tests? Please provide p values for all comparisons and justification for focus on head DNA%.

Reply: Outliers were defined as 1.5x the interquartile range (Q3-Q1), all values, including outliers, were included in the statistical processing. To evaluate the degree of DNA damage within the individual regimes of blood vessel harvest and subsequent processing, we first used Kruskal-Wallis one-way ANOVA at = 0.05 followed by the Kruskal-Wallis multiple-comparison z-value test (Dunn's test) with Bonferroni correction as described in section 2.5 Statistical analysis.

Below we present the resulting p values when comparing individual groups for Head DNA (%) and Olive moment:

Head DNA (%) Group 2 Group 4 Group 3 Group 1

Group 2 0.0000 2.2766 3.9454 2.1820

Group 4 2.2766 0.0000 0.9710 0.2639

Group 3 3.9454 0.9710 0.0000 1.4137

Group 1 2.1820 0.2639 1.137 0.0000

Olive moment Group 2 Group 4 Group 3 Group 1

Group 2 0.0000 3.6847 5.9025 1.8001

Group 4 3.6847 0.0000 1.1354 2.0295

Group 3 5.9025 1.1354 0.0000 3.8261

Group 1 1.8001 0.0295 3.8261 0.0000

Groups: (1) CAG/S - cryopreserved grafts, slowly thawed, (2) CAG/R - cryopreserved grafts, rapidly thawed, (3) FAG – cold-stored (,,fresh”) arterial grafts obtained during multiorgan harvest, and (4) CADAG - cadaveric arterial grafts harvested during autopsy.

Since the data were not normally distributed, the Kruskal-Wallis One-Way ANOVA test was used. Using Bonferroni correction medians were significantly different if z-value was > 2.6383. Values marked in red indicate all results greater than 2.6383. (p values were added into the point 3.4 of the Result section).

In our study we selected head DNA (%) as the principal parameter for the assessment of cellular damage (Table 5, Fig 4), whereas the sum of the head DNA (%) and tail DNA (%) is 100%. This parameter can be considered a sufficiently robust marker, together with tail DNA (%) (Collins 2004; Olive 2002). The other recommended parameters are tail length, tail DNA (%), or tail moment; these parameters are recommended for interlaboratory comparison of results (Kumaravel and Jha 2006; Igl 2018). (mentioned in Disscusion).

4. Table 6 indicates moderate to strong correlations between ischemia duration and DNA damage—but only in specific groups (Group 1 and Group 4). Were these associations adjusted for donor age, graft type, cold ischemia I vs II, or storage duration? Please clarify statistical models used, and whether any multivariate analyses were conducted.

Reply: We thank the reviewer for this insightful comment. The correlations presented in Table 6 (now, Fig 6) were orginally based on unadjusted analyses, as our sample size limited the feasibility of performing multivariable adjustment for potential confounders such as donor age, graft type, cold ischemia I vs. II, or storage duration. We acknowledge that these variables may have influenced the observed associations, and we have now added this point to the Discussion and Limitations section. Future studies with larger sample sizes will be needed to allow proper adjustment and to disentangle the independent contribution of ischemia duration from other covariates.

However, we added a correlation analysis to determine the association of individual comet assay parameters with donor age and storage time (performed only for grafts stored in liquid nitrogen vapor). In Group 2 (cryopreserved grafts, rapidly thawed) a strong correlation has been shown between storage duration and the following parameters: tail lenght (r = -0.616574), tail moment (r = -0.606510) and Tail area (r = -0.633326), these correlations were significant at p = 0.05000. Based on these results, it can be assumed that the level of DNA damage will increase with increasing storage time, but in group 1 (cryopreserved grafts, slowly thawed) a strong correlation was not found.

In group 1 (cryopreserved grafts, slowly thawed) an association of donor age with the parameters head DNA (%) (r = 0.69781), Tail DNA (%) (r = -0.69781), sum intensity (r = 0.647553), head radius (r = 0.663311), Olive moment (r = -0.677036) and head area (r = 0.727522) was demonstrated. These correlations were significant at p = 0.05000. In group 1, the median (Q,;Q3) age was 49 (37;56), which was, for example, a higher value than in group 2 – median (Q1;Q3) was 38 (34.5;40), however, in group 2, the association between comet assay parameters and age was not demonstrated. When performing a correlation analysis on the overall data set, no association of come assay parameters with age was demonstrated. Results of correlation analysis were added as a supplementary data (S5-S8 Table), the results were added into points 3.6 and 3.7 in the Result section.

Correlations between DNA damage rate and individual comet assay parameters were determined using Spearman's test at alfa = 0.05. The strength of a correlation was determined based on the value of the correlation coefficient according to Evans' criterion for correlation coefficient values as described in secion 2.5 Statistical analysis. Multivariate analyses were not conducted.

5. While median head DNA content > 90% suggests good DNA preservation, what thresholds are considered clinically acceptable? Is there known correlation between these comet assay metrics and graft patency, infection resistance, or long term outcomes? Please contextualize DNA damage findings relative to patient outcomes or previous benchmarks.

Reply: The median content of preserved DNA, i.e., 90% in the head of the comet, is an arbitrarily determined limit. It has been shown that DNA function, i.e., cell vitality and the ability to restart their metabolism after thawing, is closely related to this parameter (Měřička et al. 2021). In the study by Hrubý et al., 2020, the immunogenicity of cryopreserved aortic allografts was significantly lower compared to aortic allografts stored in cold storage. The correlation with graft patency and long-term results is probably due more to the extent of preservation of the endothelial layer of the vascular graft after transplantation to the recipient, as well as the preservation of the basal membrane. In such a situation, the layers of the vascular graft media that elicit a strong immune response and contribute to complications such as graft degeneration, dilation, aneurysms, and graft thrombosis in the terms of mid-term patency rates are not exposed to the recipient's immune system and are therefore directly related to the duration of its patency. Resistance to infection is determined more by the biological nature of the graft than by the preservation of its DNA and, in these terms, is not related to the high level of vital DNA preservation.

(The above text has been added to the Discussion)

6. The discussion notes there are two waiting lists in the Czech Republic (fresh vs cryopreserved), with surgeon preference guiding assignment. Can the authors provide data on wait times, rejection/discard rates, or post implant outcomes (e.g., graft-related complications), stratified by graft type and thaw protocol?

Reply: In the Czech Republic, there is a possibility to place a patient on waiting list for cryopreserved grafts since 2013. Previously, only fresh vascular grafts were available for transplantation.

Before 2013, the average waiting time for obtaining a fresh vascular graft was up to three weeks. During this time, approximately 20 % of patients did not receive a graft, with all the consequences that entailed. This was historically

---

## [Decision Letter · Decision Letter 1]

6 Oct 2025

Assessment of DNA damage of smooth muscle cells in tunica media of human arterial allografts using Comet assay method

PONE-D-25-33070R1

Dear Dr. Jandová,

We’re pleased to inform you that your manuscript has been judged scientifically suitable for publication and will be formally accepted for publication once it meets all outstanding technical requirements.

Kind regards,

Geetika Verma

Academic Editor

PLOS ONE

Additional Editor Comments (optional):

The manuscript entitled "Assessment of DNA damage of smooth muscle cells in tunica media of human arterial allografts using Comet assay method" has been reviewed and the decision of accept has been made.

Reviewers' comments:

Reviewer's Responses to Questions

**Comments to the Author**

Reviewer #1: All comments have been addressed

Reviewer #2: All comments have been addressed

2. Is the manuscript technically sound, and do the data support the conclusions?

Reviewer #1: Yes

Reviewer #2: Yes

3. Has the statistical analysis been performed appropriately and rigorously?

Reviewer #1: Yes

Reviewer #2: Yes

4. Have the authors made all data underlying the findings in their manuscript fully available?

Reviewer #1: Yes

Reviewer #2: Yes

5. Is the manuscript presented in an intelligible fashion and written in standard English?

Reviewer #1: Yes

Reviewer #2: Yes

Reviewer #1: The authors have done an excellent job revising the manuscript. They have addressed all my comments thoroughly and efficiently. The manuscript is now ready for publication, and I congratulate the authors on a well-executed and impactful study.

Reviewer #2: After careful review of the revised manuscript and the authors' responses, I find that all reviewer comments have been addressed satisfactorily. The revisions have improved the clarity, rigor, and overall quality of the work. I recommend the manuscript be accepted for publication without further modifications.

**Do you want your identity to be public for this peer review?** For information about this choice, including consent withdrawal, please see our Privacy Policy

Reviewer #1: **Yes: ** Prashant Singh

Reviewer #2: No

---

## [Editor Report · Acceptance letter]

PONE-D-25-33070R1

PLOS ONE

Dear Dr. Jandová,

I'm pleased to inform you that your manuscript has been deemed suitable for publication in PLOS ONE. Congratulations! Your manuscript is now being handed over to our production team.

Kind regards,

on behalf of

Dr. Geetika Verma

Academic Editor

PLOS ONE